# Tissue Inhibitor of Metalloproteinase 3: Unravelling Its Biological Function and Significance in Oncology

**DOI:** 10.3390/ijms25063191

**Published:** 2024-03-10

**Authors:** Wei-Ting Lee, Pei-Ying Wu, Ya-Min Cheng, Yu-Fang Huang

**Affiliations:** 1Department of Obstetrics and Gynecology, National Cheng Kung University Hospital, College of Medicine, National Cheng Kung University, Tainan 701, Taiwan; wesker1206@gmail.com (W.-T.L.); anna1002ster@gmail.com (P.-Y.W.); chengym@mail.ncku.edu.tw (Y.-M.C.); 2Department of Obstetrics and Gynecology, Kuo General Hospital, Tainan 700, Taiwan

**Keywords:** tissue inhibitor of metalloproteinases-3, gynaecological cancers, biomarker, cancer therapy

## Abstract

Tissue inhibitor of metalloproteinases-3 (TIMP3) is vital in regulating several biological processes. TIMP3 exerts antitumour effects via matrix metalloproteinase (MMP)-dependent and MMP-independent pathways. Due to promoter methylation and miRNA binding, TIMP3 expression has been observed to decrease in various cancers. Consequently, the migration and invasion of cancer cells increases. Conflicting results have reported that expression levels of TIMP3 in primary and advanced cancers are higher than those in healthy tissues. Therefore, the role of TIMP3 in cancer biology and progression needs to be elucidated. This review provides an overview of TIMP3, from its biological function to its effects on various cancers. Moreover, gynaecological cancers are discussed in detail. TIMP3 has been associated with cervical adenocarcinoma as well as cancer development in serous ovarian cancer and breast cancer metastasis. However, the relationship between TIMP3 and endometrial cancers remains unclear. TIMP3 may be a useful biomarker for gynaecological cancers and is a potential target for future cancer therapy.

## 1. Introduction

In 2020, an estimated 10 million deaths from cancer were observed worldwide [1]. Numerous factors may contribute to the increased risk of developing cancer, including exposure to environmental hormones and pollution, smoking, unhealthy diet, infection, and ageing [2,3,4]. To identify the risk factors for cancer occurrence, predictors for early diagnosis and individualised and effective treatment are crucial for comprehensive cancer control. Developing effective screening strategies for asymptomatic cancer patients could decrease the incidence of late-stage cancer and increase the effectiveness of cancer treatment. However, the early detection of asymptomatic and developing cancers is challenging [5,6]. We are confronted with the formidable challenges of metastasis and chemoresistance, which represent crucial impediments in the landscape of cancer therapy [7,8]. Identifying the proteins involved in metastasis and chemoresistance would help identify potential strategies against cancer.

Despite developing advancements in cancer management, cancer remains a crucial public health and economic issue. For instance, epithelial ovarian cancer (EOC) is affecting the lives of women in Asia as the number of new cases increases [9,10]. The combination of surgical cytoreduction and chemotherapy (e.g., platinum and paclitaxel) or radiotherapy, as the standard treatment for gynaecological cancer [11,12,13], in conjunction with targeted agents or immune checkpoint inhibitors, has been implemented in the clinic. Bevacizumab, a recombinant humanised monoclonal antibody against vascular endothelial growth factor (VEGF), has been reported to improve progression-free survival (PFS) in patients with gynaecological cancers [8,14,15] and overall survival (OS) in high-risk EOC populations [14,15] as well as patients with cervical cancer [16]. Pembrolizumab, a humanised monoclonal antibody against the programmed cell death protein 1 (PD-1) receptor, has been demonstrated to extend the median progression-free survival (PFS) compared to chemotherapy alone in endometrial cancer [17]. Olaparib, a poly-ADP ribose polymerase (PARP) inhibitor, has been reported to extend the PFS and OS in patients with BRCA-mutated EOC [18]. Breakthrough evidence has shown that antibody–drug conjugates targeting folic acid receptor alpha (FRα) are overexpressed in gynaecological cancers [19]. Mirvetuximab soravtansine (MIRV), an antibody–drug conjugate (ADC) drug against folate receptor α (FRα), was approved by the U.S. Food and Drug Administration (FDA) for FRα-positive platinum-resistant EOC because MIRV successfully prolonged PFS and OS [20]. 

Tissue inhibitor of metalloproteinases-3 (TIMP3) have been demonstrated to suppress cancer progression in vitro and in vivo. The validity of this claim is still being discussed in academic circles due to varying and inconclusive findings. This study examines the role of TIMP3 in oncology, particularly in gynaecological cancers. Moreover, the biological function of TIMP3 and its effects on cancer through its regulation of TIMP3 or related molecules are discussed in this study. 

## 2. TIMP3 Biology

TIMP3, a member of the tissue inhibitors of the metalloproteinase family, is approximately 24 kDa in size, and its glycosylated TIMP3 is approximately 27 kDa [21]. Both endogenous and exogenous molecules regulate TIMP3 expression. Leivonen et al. indicated that transforming growth factor β-1 (TGF-β1) induced TIMP3 gene expression in normal human gingival fibroblasts via Smad3/Smad4 signalling [22]. The same study indicated that p38, ERK1/2, and Smad3 synergistically mediated the upregulation of TIMP3 expression [22]. TIMP3 is known for a critical role in the regulation of extracellular matrix (ECM) stability through inhibiting various matrix metalloproteinases (MMPs), A disintegrin and metalloproteases (ADAMs) as well as ADAM with thrombospondin motifs (ADAMTSs) [23]. TIMP3 binds with 1:1 stoichiometry to the target, inhibiting its activity [24]. An imbalance between TIMP3 and MMPs/ADAMs/ADAMTSs causes various diseases, including myocardial infarction, Alzheimer’s disease, intervertebral disc degeneration, impaired cognitive function, and tumour metastasis [25,26,27,28,29]. TIMP3 has been demonstrated to regulate inflammation through the inhibition of ADAM17, which indirectly decreases TNF [30]. TIMP3 promotes endothelial cell apoptosis by inhibiting matrix-mediated tyrosine phosphorylation of FAK [31]. TIMP3 has been demonstrated to inhibit cell proliferation and migration by inhibiting MMP-2 and MMP-3 inhibition [32]. TIMP3 suppresses neuronal differentiation by upregulating Notch signalling and suppressing MMPs in neural stem cells [33]. Moreover, it exerts anti-angiogenic effects by inhibiting cell proliferation and migration through inhibition of MMPs and interference with the binding of vascular endothelial growth factor (VEGF) and VEGF receptor 2 (VEGFR2) [34,35]. Cruz et al. detected a hypomethylated TIMP3 promoter in the placental samples from patients with preeclampsia [36]. 

Furthermore, TIMP3 has been identified as a candidate biomarker for several diseases, including diabetic nephropathy, myocardial infarction, and cancer progression [37,38,39].

## 3. Regulation of TIMP3

Various studies showed that the TIMP3 level in different cancer types was regulated by other molecules (Figure 1), characterised by organ-specific gene expressions.

### 3.1. Upregulation of TIMP3

The molecules that upregulate TIMP3 expression are listed in Table 1. Oncostatin M, an Interleukin 6 (IL-6) family member, strongly activates TIMP3 mRNA expression in bovine chondrocytes [40]. Gatsios et al. found that oncostatin M decreased expression of TIMP3 mRNA in human synovial lining cells, while IL-1β upregulated TIMP3 mRNA level [41]. The expression level of TIMP3 mRNA was reduced in murine brain microvascular endothelial cells and rat astrocytes after treatment with IL-1β/tumour necrosis factor-alpha (TNFα) and Interferon gamma (IFNγ)/TNFα, respectively [42]. The discordant results of the studies above may be due to using different cell lines. Leco et al. showed that the mRNA level of TIMP3 was upregulated by TGF-β1, phorbol ester, dexamethasone, and epidermal growth factor (EGF) in mouse fibroblast [43]. Exogenous IL-27 upregulates TIMP3 mRNA expression in prostate cancer cells [44]. MPT0G013, an arylsulfonamide-based derivative, upregulates TIMP3 in HUVEC and colon cancer cells in mouse xenograft models [45]. Another aryl sulfonamide-based derivative, MPT0B390, transcriptionally upregulates TIMP3 levels in colon cancer cell lines and HUVEC by inhibiting the expression of the enhancer of zest homolog 2 (EZH2, a histone methyltransferase) [46]. 3-Deazaneplanocin A (DZNep), an EZH2 inhibitor, increased TIMP3 mRNA levels in liver cancer cells [47]. AG014699 and BSI-201, which are PARP-1 inhibitors, increased TIMP3 protein levels in hepatocellular carcinoma cells [48]. Nuclear factor erythroid 2-related factor 2 (Nrf2) increased TIMP3 expression in mouse hepatic macrophages [49]. miR-29c upregulated TIMP3 expression in breast cancer cells by downregulating DNA methyltransferase 3B (DMNT3B, a DNA methyltransferase) [50].

### 3.2. Downregulation of TIMP3

The molecules that downregulate TIMP3 expression are listed in Table 2. Human proinsulin-connecting peptide (C-peptide), produced by pancreatic β-cell, downregulates TIMP3 gene expression and upregulates MMP9 in human endometrial stromal cells via a β-catenin-dependent pathway, which contributes to cellular migration and invasion [53]. With respect to TIMP3 downregulation, miRNA targeting and promoter methylation are the determinant factors. TIMP3 has the extended 3′-untranslated region (UTR) contained in the exon 5; hence, multiple miRNAs can target it, leading to the degradation of TIMP3 mRNA [54]. miR-21-targeted TIMP3 leads to decreased TIMP3 expression and upregulates MMP2 and MMP9, enhancing capillary network formation in HUVEC cells [55]. Additionally, exosomal miR-17-3p reduces the number of necrotic cardiomyocytes by negatively regulating the expression of TIMP3, which promotes H_2_O_2_-induced programmed necrosis in primary cardiomyocytes [56]. miR-34b-5p is associated with bleomycin-induced pulmonary fibrosis by decreasing TIMP3 expression [57]. In colon cancer cells growing in the liver, cyclin-dependent kinase 8 reduces TIMP3 expression by inducing miR-181b [58]. In contrast, miR-136 inhibits TIMP3 and protects neurocytes from hypoxia-induced apoptosis [59]. The high-mobility group box-1 protein downregulates TIMP3 through upregulating miR-206, which is involved in improving myocardial regeneration, angiogenesis, and collagenolytic activity in failing hearts [60]. Furthermore, Su et al. provided an overview of various miRNAs that target TIMP3 in cancer cells [61]. Methylation of tumour suppressor genes is a common phenomenon in cancer, which silences genes and contributes to cancer progression. TIMP3 promoter methylation is often found in various cancers, including oral, gastric, and cervical cancer [62,63,64]. 

## 4. TIMP3 in Non-Gynaecological Cancers

The expression levels of TIMP3 in different non-gynaecological cancers are shown in Table 3. TIMP3 has been described as a tumour suppressor in several human malignancies, including liver, lung, thyroid, colon, and head and neck cancer. Cancer patients with decreased TIMP3 expression have poor outcomes [65,82,83,84]. The antitumour effects of TIMP3 depend on both MMP-dependent and MMP-independent pathways. MMPs released from tumour and stromal cells play vital roles in dynamic ECM processing, angiogenesis, and immune escape [85]. Cancer cells invade adjacent tissues and spread to other locations thereafter via hematogenous or lymphatic spread [86]. Reduced TIMP3 expression is thought to result from aberrant promoter hypermethylation [87,88] and miRNA regulation in several tumour types [89,90].

### 4.1. Lung Cancer

Lung cancer is the most common cancer in the world. TIMP3 is considered an essential molecule in lung cancer. TIMP3 expression levels in different tumour stages are downregulated compared to those in the normal tissue group [66]. Mino et al. showed that patients with low TIMP3 expression exhibited increased nodal metastases and poor 5-year OS rates [91]. KDM1A directly decreases TIMP3 promoter activity, improving lung cancer progression and unfavourable outcomes [65]. The downregulation of TIMP3 expression by miRNA-197-3p promotes angiogenesis [72]. Krüppel-like factor 4 (KLF4) is a transcription factor that facilitates the transcription of the tumour suppressor TIMP3 by directly binding to the TIMP3 promoter, inhibiting cancer progression in vitro and in vivo [51]. IL-32γ reduces lung cancer cell growth in vitro and in vivo by inhibiting the binding of NFκB-dependent DNA (cytosine-5)-methyltransferase 1 (DNMT1) to TIMP3 promoter and thus increased TIMP3 expression contributes to the inhibition of cancer growth [52].

### 4.2. Head and Neck Cancer

Su et al. showed that the mean plasma TIMP3 level was lower in oral squamous cell carcinoma (OSCC) than in healthy controls [39]. TIMP3 levels in the plasma of patients with OSCC are significantly associated with tumour status; however, they are not associated with lymph node status, metastasis, or cell differentiation [39]. Dressing TIMP3 by DNA methylation contributes to oral cancer metastasis [83,92]. miR-221 has been reported to increase OSCC resistance to adriamycin and doxorubicin via TIMP3 inhibition [73,74].

Nevertheless, The Cancer Genome Atlas (TCGA) data show that the mRNA levels of TIMP3 do not differ between OSCC tissues and normal tissues [39]. Kornfeld et al. have demonstrated higher TIMP3 mRNA levels in the stroma of head and neck cancer cells than those in normal epithelial cells [93]. Clinically, patients with higher levels of TIMP3 mRNA in tumour-associated stromal areas have unfavourable clinical outcomes [93].

### 4.3. Liver Cancer

The TIMP3 mRNA level in liver cancer tissues is lower than in paired adjacent non-cancerous tissues [94]. The positive TIMP3 has been correlated with less portal vein invasion, nodal metastasis, better PFS and OS [94]. The upregulation of TIMP3 by DZNep is associated with attenuating proliferation of liver cancer cells and an increased population of apoptotic cells [47]. miR-181b decreases TIMP3 expression and promotes the tumourigenic properties of liver cancer cells in vitro and in vivo [70].

### 4.4. Colorectal Cancer

In colorectal cancer (especially rectal cancer), patients with high-cytoplasmic-staining levels of TIMP3 have a comparatively high 5-year survival rate [95]. Powe et al. found that the mRNA signals of TIMP3 were frequently detected in the invasive edge of moderately and poorly differentiated colorectal adenocarcinoma samples compared with well-differentiated carcinomas and paired distant stroma tissues [96]. The lack of TIMP3 may enhance the invasion ability of poorly differentiated tumours [96]. By upregulating TIMP3 expression, MPT0G013 and MPT0B390 suppress the proliferation and metastasis of colon cancer tumours in vitro and in vivo [45,46]. Although MPT0B390 has been developed for cancer therapy, its effects on gynaecological cancers have not yet been evaluated. TIMP3 also decreases the levels of CD44 and reduces the motility of colorectal cells [97]. CircFNDC3B, a circular RNA that sequesters miR-937-5p, leads to elevated expression levels of TIMP3 and inhibits mouse colorectal cancer progression [80]. Conversely, Konishi et al. indicated that the percentage of TIMP3 methylation was lower in primary colorectal cancer with liver metastases than in primary cancer without liver metastases [98]. 

### 4.5. Thyroid Cancer

The methylation of TIMP3 has commonly been found in thyroid cancer tissues and associated with extrathyroidal invasion and lymph node metastasis [99]. Yang et al. showed TIMP3 is inversely correlated with miR-221/222, and the aggressive thyroid cancer tissues have relatively low TIMP3 mRNA levels compared to non-aggressive cancer tissues [75]. According to TCGA data, the TIMP3 expression level is lower in thyroid cancer tissues than in normal tissues and is correlated with the OS of thyroid cancer patients [100]. TIMP3 reduces thyroid cancer cell proliferation, migration, and invasion of thyroid cancer [82]. Baldini et al. found that the TIMP3 mRNA signal in anaplastic thyroid carcinoma-derived cell lines (CAL-62 and 8305C) is lost [101]. Based on Anania and Baldini’s results, different cell lines show various TIMP3 mRNA levels, although these cell lines have the same histology of thyroid carcinomas [82,101].

### 4.6. Bone Cancer

Guo et al. found that the expression of TIMP3 mRNA significantly decreases in human osteosarcoma tissues compared to that in matched adjacent normal tissues [78]. The authors also found an inverse correlation between TIMP3 and miR-222-3p expression levels. Downregulation of TIMP3 by miR-222-3p increases proliferation and osteosarcoma cell metastasis [78]. TIMP3 overexpression improves the sensitivity of osteosarcoma cells to cisplatin by inhibiting AKT activation and IL-6 production [102].

### 4.7. Gastric Cancer

The promoter methylation of TIMP3 has been detected in gastric carcinoma [103]. The percentage of TIMP3 promoter methylation increases significantly among early, advanced, and metastatic gastric cancer tissues compared to normal tissues [63]. George et al. indicated that TIMP3 is consistently downregulated and hypermethylated in gastric cancer and gastric stomach tissues of Helicobacter pylori-infected patients [104]. TIMP3 methylation correlates with lymph node metastasis in patients with gastric cancer but not with OS [105]. Li et al. showed that the expression levels of TIMP3 are higher in normal tissue than in gastric cancer tissue [106]. However, they found that gastric cancer patients with relatively high levels of TIMP3 have unfavourable OS. Their results demonstrated that gastric cancer cells become less aggressive after the downregulation of TIMP3 [106].

### 4.8. Breast Cancer

Breast cancer is the most common cancer in women. Some reports show that TIMP3 plays a vital role in breast cancer. Breast cancer patients with relatively high levels of TIMP3 mRNA have longer disease-free survival (DFS) and better responses to tamoxifen [107,108]. Bi et al. reported that the RNA-binding protein Musashi1 (Msi1) is upregulated, and TIMP3 is downregulated in metastatic breast cancer [109]. Mechanistically, Msi1 is physically bound to 3′UTR of TIMP3, which results in TIMP3 suppression and then MMP9 upregulation [109]. The downregulation of miR-21 increases the expression level of TIMP3 and decreases cell invasion [67]. Reduced TIMP3 expression and increased CD44 expression strongly correlate with nodal involvement in breast cancer patients [110]. Downregulation of miR-221/222 correlates with increased TIMP3 expression and the sensitivity of MCF-7 breast cancer cells to tamoxifen [77]. Işeri et al. found that TIMP3 is downregulated in docetaxel- and doxorubicin-resistant MCF-7 cell lines compared to sensitive counterparts [111]. Moreover, primary breast tumour tissue shows a significantly higher proportion of TIMP3 methylation than matched normal tissue [112]. Zhou et al. showed that TIMP3 is activated by a high-mobility group (HMG) box-containing protein 1 (HBP1) and then stabilised by phosphatase and tensin homolog (PTEN) [113]. Consequently, breast cancer becomes sensitive to radiation and hormonal therapies [113]. TIMP3 may be a useful biomarker for breast cancer prognosis and drug response. Nevertheless, transgenic mice deficient in TIMP3 show delayed breast tumour development, progression, and decreased incidence [114].

Various reports have been published regarding the effects of TIMP3 on other types of cancer. The expression levels of TIMP3 in oesophagal cancer tissues and plasma from patients are significantly lower than those in normal tissue and plasma from healthy volunteers, respectively [79]. The downregulation of TIMP3 by miR-373 also increases the proliferation of oesophagal cancer cells and metastatic ability [79]. Shen et al. noted that TIMP3 levels are lower in cisplatin-resistant laryngeal carcinoma tissues and that patients with common TIMP3 expression have unfavourable OS [115]. TIMP3 has been shown to increase prostate cancer cell sensitivity to paclitaxel via mitochondrion-mediated caspase-3 activation [116]. miR-21 has been shown to target TIMP3, is upregulated in various solid and haematological malignancies, and is linked to high cell proliferation, high invasion, anti-apoptosis, and metastatic potential by targeting the expression of several genes [117].

Although TIMP3 has tumour-suppressive potential, some studies have suggested that TIMP3 promotes carcinogenesis [92,93]. Increased TIMP3 signal intensity has been detected in pancreatic cancer tissues; however, the signal is weak in normal tissues [118]. Moreover, upregulated TIMP3 is associated with Thrombospondin 1 (THBS1) in the protein-protein interaction network, and upregulated TIMP3 may be involved in the ECM–receptor interaction signalling pathway during cancer metastasis [119,120]. 

**Table 3 ijms-25-03191-t003:** Studies report TIMP3 expression levels in non-gynaecological cancers.

Cancer Type	Reference(s)	Sub-Group	Case No.	TIMP3 Level	Method
Lung cancer	[65]	Normal	59	High	RNA-seq (TCGA database)
		T1	170	Low	
		T2	278	Low	
		T3	47	Low	
		T4	19	Low	
	[91]	Normal	87 (paired)	High	IHC analysis
		Cancer	92	Low	
Head and neck cancer	[39]	Normal	64	11,289.9 ± 952.1 ^#^	ELISA (Plasma)
		Cancer	450	3845.0 ± 167.8 ^#^	
	[83]	Normal	17 (paired)	High	Q-PCR
		Cancer	17	Low	
		Normal	8 (paired)	High	Western blot
		Cancer	8	Low	
Liver cancer	[94]	Normal	20 (paired)	High	Q-PCR
		Cancer	20	Low	
Colorectal cancer	[46]	Normal	159	High	GEPIA
		Cancer	257	Low	
		Normal	3	High	IHC analysis
		Cancer	3	Low	
Thyroid cancer	[82]	Normal	9	High	cDNA microarray
		Classical	21	Low	
		Tall cell	10	Lowest	
	[75]	Non-aggressive Aggressive	20	High	Q-PCR

			20	Low	
Bone cancer	[78]	Normal	30 (paired)	High	Q-PCR
		Cancer	30	Low	
	[102]	Cisplatin sensitive Cisplatin resistant	4	High	IHC analysis

			4	Low	
Breast cancer	[108]	Normal	17	High	IHC analysis
		Metastatic	104	Low	
Oesophagal cancer	[79]	Normal	63 (paired)	High	Q-PCR (Tissues)
		Cancer	63	Low	
	[79]	Normal	39	High	Q-PCR (Plasma)
		Cancer	63	Low	
Pancreatic cancer	[118]	Normal	10	8 (80.0%) positive, low	IHC analysis
		Cancer	75	55 (73.3%) positive, high	

Only Su et al. [61] reported the number of ELISA results and showed the cutoff value. The results of IHC, Q-PCR, Western blotting, and array analyses did not show the number in the research reports. GEPIA, gene expression profiling interactive analysis; IHC, immunohistochemistry; Q-PCR, quantitative polymerase chain reaction; ELISA, enzyme-linked immunosorbent assay; ^#^ mean ± standard deviation, the unit is pg/mL.

## 5. TIMP3 in Gynaecological Cancers

These studies have demonstrated the effects of TIMP3 on cancers through different mechanisms; however, discussions of TIMP3 in gynaecological cancers are rare. Table 4 shows the expression levels of TIMP3 in various gynaecological cancers.

### 5.1. Cervical Cancer

The incidence of cervical cancer has declined since the advancement of screening strategies and the worldwide promotion of papillomavirus (HPV) vaccination. HPV vaccination and cervical cancer screening can effectively help reduce the risk of contracting cervical cancer and improve the likelihood of finding cervical cancer at an early stage, respectively [121,122]. Squamous cell carcinoma (SCC) and adenocarcinoma are the most common types of cervical cancer [123]. Previous studies have shown that patients with adenocarcinoma have worse OS than those with SCC [124,125,126].

The proportion of methylated TIMP3 in cervical cancer is significantly higher than that in normal cervical tissue [127,128]. Studies have reported that TIMP3 was more frequently methylated in cervical adenocarcinoma than in SCC (53.3–63.0% vs. 5.0–8.1%) [128,129,130]. However, Siegel et al. reported no significant differences in the TIMP3 methylation index between SCC and normal tissues [130]. Based on previous studies, TIMP3 methylation is a potential biomarker to distinguish cervical adenocarcinoma from SCC.

Some studies have indicated that the expression level of TIMP3 is lower in cervical intraepithelial neoplasia and cancer tissues than in normal samples [131,132,133,134] (Table 4). According to the TCGA data, TIMP3 expression is lower in cancer samples than in normal samples [131]. Compared to patients with lower expression levels of TIMP3 in cervical cancer tissues, those with higher TIMP3 expression correlate with a lower survival rate; however, no significance has been observed [131].

An inverse correlation between miR-21 and TIMP3 expression has been demonstrated in cervical cancer samples [134]. Moreover, Shishodia et al. reported that the expression level of miR-21 increases during the transition from low-grade squamous intraepithelial lesions (LSIL) to high-grade squamous intraepithelial lesions (HSIL) and invasive cancer, corresponding to a decreased level of TIMP3 [68]. miR-221/222 have been demonstrated to target the 3′ untranslated regions (UTR) of TIMP3 in cervical cancer and lead to an increase in the levels of MMP2 and MMP9, as well as the promotion of cell migration and invasion in cervical cancer [76]. miR-G-10 represses TIMP3 expression, preventing the increased migration and invasiveness of cervical cancer cells [81].

**Table 4 ijms-25-03191-t004:** The expression levels of TIMP3 in different gynaecological cancerous tissues.

Cancer Type	Reference(s)	Sub-Group	Cases No.	TIMP3 Level	Method
Cervical cancer	[130]	Normal	33	High	Q-PCR
		CIN	23	Low	
		CC	8	Lowest	
	[134]	Normal Adjacent non-neoplastic CC	40 (Paired)	Highest	Q-PCR and Western blot
			40	High	
			40	Lowest	
		Normal	3	High	Q-PCR (TCGA database)
		CESC	305	Low	
	[132]	Normal	3	High	cDNA arrays
		CC	3	Low	
	[135]	LSIL	12	Weak	IHC analysis
		HSIL	11	Moderate to strong	
		ISCC	8	Strong	
Ovarian cancer	[136]	Simple cysts	30	285 (148–368) *	MFBBI (serum)
		Endometrial	30	223 (143–276) *	
		ovarian cysts			
		Serous ovarian cancer	44	138 (67–198) *	
	[71]	Healthy controls	12	High	Q-PCR and Western blot
		Endometriomas	12	Low	
		EAOC	12	Lowest	
	[137]	Primary	419	Low	mRNA microarray (3 public datasets)
		Metastatic SOC	145	High	
		Normal	8	Low	Q-PCR
		Primary SOC	30	High	
		Metastatic SOC	29	Highest	
	[138]	Benign	9	Low	IHC analysis
		Borderline	9	High	
		Malignant	28	Highest	
	[139]	Normal	22	0.13 ± 0.67 ^#,$^	Q-PCR
		Benign	21	0.85 ± 0.75 ^#,$^	
		Malignant	60	0.87 ± 0.46 ^#,$^	
	[140]	Normal	3	Low	Q-PCR
		HGSOC	3	High	
	[141]	Benign	8	Low	IHC analysis
		SOC	26	High	
Endometrial cancer	[142]	Benign	1	High	cDNA expression array
		WDEAC	2	Low	
	[143]	Adenocarcinoma	27	Strong	IHC analysis
		Squamous	1	Strong	
		Clear cell	1	Strong	

Cymbaluk-Płoska et al. [136] and Hu et al. [139] presented interval data, whereas other studies have not shown quantitative data of IHC, Q-PCR, Western blotting, and array. CIN, cervical intraepithelial neoplasia; CC, cervical cancer; CESC, cervical squamous cell carcinoma; LSIL, low-grade squamous intraepithelial lesions; HSIL, high-grade squamous intraepithelial lesions; ISCC, invasive squamous cell carcinoma; Q-PCR, quantitative polymerase chain reaction; IHC, immunohistochemistry; MFBBI, multiplex fluorescent bead-based immunoassays; SOC, serous ovarian cancer; EAOC, endometriosis-associated ovarian cancer (8 endometrioid and 4 clear cell tumour); HGSOC, high-grade serous ovarian cancer; * mean of the range, the unit is pg/mL; ^#^ mean ± standard deviation. ^$^ The ratios of TIMP3/β-actin were used to represent the semiquantitative expression of TIMP3.

A study with contrasting findings has indicated that TIMP3 is upregulated in cervical cancer cell lines (HPV-related QG-U cells and HPV-negative Yumoto cells) and cervical SCC tissues [135]. Shaker et al. created a model to induce LSIL, HSIL, and invasive cervical cancer by introducing genetic material into human cervical keratinocytes (HCK). They noticed that the expression levels of TIMP3 increase during the carcinogenic process in normal cervical cells compared with parental HCK cells. Furthermore, the strong immunoreactivity of TIMP3 has been detected in both nuclear and cytoplasmic patterns in HSIL and invasive cervical cancer tissues [135].

The differences between these findings may be due to different detection methods for TIMP3 and samples. However, further studies are required to confirm the role of TIMP3 in cervical carcinogenesis and cancer progression.

### 5.2. Epithelial Ovarian Cancer (EOC)

EOC is a silent killer that is difficult to recognise in its early stages and poses a global threat to women. Patients with advanced-stage EOC had significantly worse OS than those with the early-stage disease. In the United States, ovarian cancer is the fifth leading cause of death in female malignancies, and those with distant metastases have the worst survival rates [144]. EOC is a heterogeneous disease that manifests histologically, molecularly, and clinically. Serous carcinoma is the most common histology worldwide [145,146], whereas the proportion of clear-cell carcinomas (CCC) is relatively higher (15–20%) in Asian countries [145,147,148].

The mean concentration of the TIMP3 protein in the sera of patients with EOC is significantly lower than that in patients with benign ovarian cysts or endometrial cysts [136]. Higher TIMP3 levels are inversely correlated with ascites in patients with advanced stages of ovarian cancer [119]. Dong et al. showed that the expression level of TIMP3 is inversely correlated with miR-191 expression [71]. The expression level of TIMP3 is the highest, and miR-191 lowest in healthy control samples [71]. Patients with higher TIMP3 have an 8.9-month increase in OS compared with those with lower TIMP3 levels [136]. Hakamy et al. indicated that patients with EOC with relatively high TIMP3 expression have prolonged disease-specific survival compared to patients with common TIMP3 expression [149]. TIMP3 can be induced by physcion 8-O-β-glucopyranoside (PG) and lead to the suppression of the migration and invasion of serous EOC cells [150]. Moreover, PG induces other anti-cancer molecules. However, the precise mechanism by which TIMP3 is directly upregulated by PG is not well understood. Silencing Snail expression increases the expression level of TIMP3, leading to decreased proteolytic activity of MMP2 and MMP9, whereas normalised expression of Snail inhibits TIMP3 expression, resulting in increased activity of MMP2 and MMP9 in EOC [151]. The expression level of the TIMP3 gene can be induced in both ovarian stromal cells and cancer cells by TGFβ-1, which regulates cell proliferation, migration, and differentiation [137]. These results were consistent with the role of TIMP3 as a cancer suppressor, as mentioned earlier.

TIMP3 gene expression dynamics have been observed during the dormancy-to-recurrence transition induced by VEGF/doxycycline (DOX) or DOX in vitro and in vivo models, respectively [120]. When serous EOC cell dormancy was induced with DOX, TIMP3 expression increased in the cancer cells. After the withdrawal of DOX, the EOC cells underwent recurrent growth and an increase in the TIMP3 expression was detected in the DNA methylation [120]. In addition, the mRNA and protein levels of TIMP3 were higher in carboplatin/paclitaxel-induced senescent primary serous EOC cells than in young cells [152]. Dormancy and senescence are critical cellular stress responses that contribute to therapy resistance and tumour recurrence [153]. Thus, TIMP3 may be a key molecule in cancer cell dormancy and senescence. 

Controversial results have reported that TIMP3 tends to be highly expressed in EOC patients at higher pathological stages [137,151,152,153,154,155]. Januchowski et al. reported that higher expression levels of TIMP3 were detected in cisplatin-resistant A2780 cell lines than that in their sensitive counterparts [156]. Cheon et al. showed an inverse correlation between TIMP3 expression and unfavourable outcomes in patients with serous EOC [137]. Furthermore, they also found that TIMP3 was highly enriched in metastatic tissues compared to that in primary tumours [137]. Lima et al. reported strong immunoreactive signals of TIMP3 in the EOC group compared to borderline and benign neoplasms [138]. Their patients with EOC and higher TIMP3 expression had shorter OS (94.5 months vs. 156.2 months) [138]. Hu et al. showed that malignant and benign tissues express higher levels of TIMP3 than to normal tissues [139]. Zhang et al. reported that the expression levels of TIMP3 mRNA are significantly upregulated in serous EOC samples compared with those in normal ovaries [140]. The protein levels of TIMP3 in the culture supernatants of TOV-21G (CCC histology) and TOV-112D (endometrioid histology) cells, both of which are from grade 3 tumours, were detected using a guided-mode resonance (GMR) bioassay detection system [154]. Significant upregulation of TIMP3 expression was observed in FIGO Stage III and Grade 3 EOC; however, it was not found in benign ovarian samples [141]. Methylated TIMP3 has been identified in various cancers. However, Imura et al. showed that partial methylation of TIMP3 was observed in two EOC cell lines, and the remaining 11 EOC cell lines (different types) exhibited TIMP3 demethylation [157]. TIMP3 has been identified as an ovarian cancer-specific biomarker of cancer-associated fibroblasts (CAFs) associated with cancer progression, chemoresistance, and poor prognosis by integrating several bioinformatic approaches [155]. 

Based on previous reports, the expression levels of TIMP3 in most serous EOC tissues are higher than in normal tissues. However, the correlation between TIMP3 expression and cancer progression in EOC remains inconsistent. Owing to the lack of investigation into TIMP3 expression in ovarian CCC, further studies are necessary.

### 5.3. Endometrial Cancer

Most endometrial cancers are diagnosed at the early stage (I and II) in postmenopausal women with abnormal uterine bleeding [142]. New endometrial cancer cases have increased worldwide since 2020 [144,145]. Few studies have focused on the role of TIMP3 in endometrial development.

Smid-Koopman et al. revealed that TIMP3 expression was downregulated in two well-differentiated endometrioid adenocarcinoma samples compared to benign human endometrial tissue [158] (Table 4). miR-103 and miR-181a have been demonstrated to repress the expression levels of TIMP3 through directly binding to TIMP3’s 3′-UTR in Ishikawa cells and HEC-1B cells [69,90]. Yu et al. reported that the proliferation and invasiveness of endometrial cancer cells are improved after transfection with anti-miR-103 [69]. In contrast, Di Nezza et al. found that all endometrial carcinoma tissues of all histological grades show strong immunoreactivity for TIMP3, and myometrial invasion is present in 78% of patients [143]. Further studies with larger sample sizes are required to elucidate the role of TIMP3 in endometrial cancer.

## 6. Conclusions and Perspectives

TIMP3 is important in ECM remodelling and involves inflammation, cardiovascular diseases, neurological disorders, and cancer progression. Most studies have shown that TIMP3 is involved in tumour inhibition in most cancer types. Higher TIMP3 levels have been detected in normal human tissues than in cancerous tissues. Cancer patients with lower TIMP3 levels have less favourable outcomes. Nevertheless, controversial results have been observed in gastric, pancreatic, cervical, and ovarian cancer. This overview provides fundamental knowledge of the biological functions of TIMP3 in cancer cells (Figure 2) and summarises the mechanical interactions between different cancers and TIMP3. However, the correlations between vaginal or vulvar cancer and TIMP3 are not included in this review because of a lack of relevant studies. The expression levels of TIMP3 in cervical cancer are lower based on in vitro and clinical studies. The expression levels of TIMP3 in serous EOC tissues are higher than those in healthy tissues; however, the underlying mechanism remains unclear. Only one study with a larger sample size (*n* = 29) indicated that higher expression levels of TIMP3 had been detected in endometrial cancer tissues [143]. TIMP3 may act as a biomarker of cancer prognosis and drug response. Reports have suggested that the threshold or cut-off value of TIMP3 levels and concentrations vary across different types of cancer. Therefore, the upcoming issues are standardising detection methods and validating the findings in more clinical samples. 

As mentioned earlier, TIMP3 has classically been considered a tumour suppressor protein. Therefore, stimulating TIMP3 expression in cancer cells may enhance therapeutic efficacy. For instance, TIMP3 may be upregulated by molecules (e.g., IL-27, DZNep and MPT0B390) or compounds from natural products. Green tea polyphenols and epigallocatechin-3-gallate elevate TIMP-3 expression by reducing the protein levels of EZH2 and class I histone deacetylases, which attenuate the migration of breast cancer cells [159]. KHBJ-9B, a butanol fraction extracted from a mixture of two oriental herbs, increases TIMP3 levels and decreases the expression of matrix proteinases in human osteoarthritic cartilage cultures [160]. The crude acetone extract of *Momordica balsamina* increased TIMP3 expression in colorectal cancer, reducing migration ability [161]. Purified molecules or natural products can be used as adjuvants for cancer therapy.

Nanotechnology has advanced applications in cancer diagnosis and therapy [162,163]. Nanoparticles can be modified with specific targeting molecules and loaded with specific chemicals to improve their therapeutic efficacy [164,165]. Zhou et al. developed a multifunctional nanoparticle that co-delivered a miR-221/222 inhibitor and paclitaxel to MDA-MB-231 breast cancer cells [166]. The expression levels of p27Kip1 and TIMP3 are upregulated in cancer cells due to the inhibition of miR-221/222, which enhances the therapeutic efficacy of paclitaxel [166]. Li et al. developed copper-olsalazine (Cu-Olsa)@hyaluronic acid (HA) nanoparticles that caused COX-2 downregulation, reactive oxygen species generation, and TIMP3 upregulation in colorectal cancer cells [166]. Cu-Olsa@HA nanoparticles significantly inhibited colorectal cancer proliferation and metastasis in vitro and in vivo [167]. Hence, developing smart nanoparticles targeting TIMP3-related processes may be a strategy for efficiently treating various cancers.

## Figures and Tables

**Figure 1 ijms-25-03191-f001:**
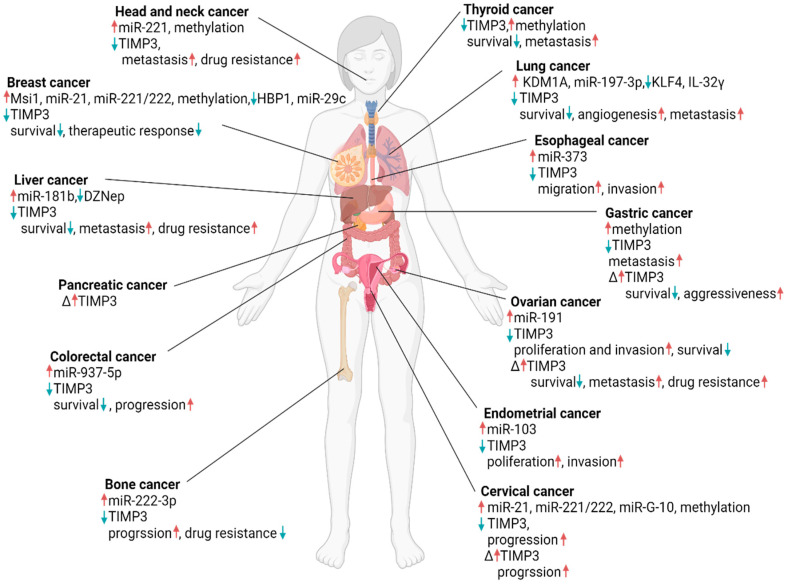
The influence of different molecules on tissue inhibitor of metalloproteinases-3 (TIMP3) regulation in various cancers and its impact on patient outcomes. The symbol “**↑**” indicates upregulation, and the symbol “**↓**” indicates downregulation. The symbol “∆” indicates the controversial result. This was created with BioRender.com (accessed on 1 March 2024).

**Figure 2 ijms-25-03191-f002:**
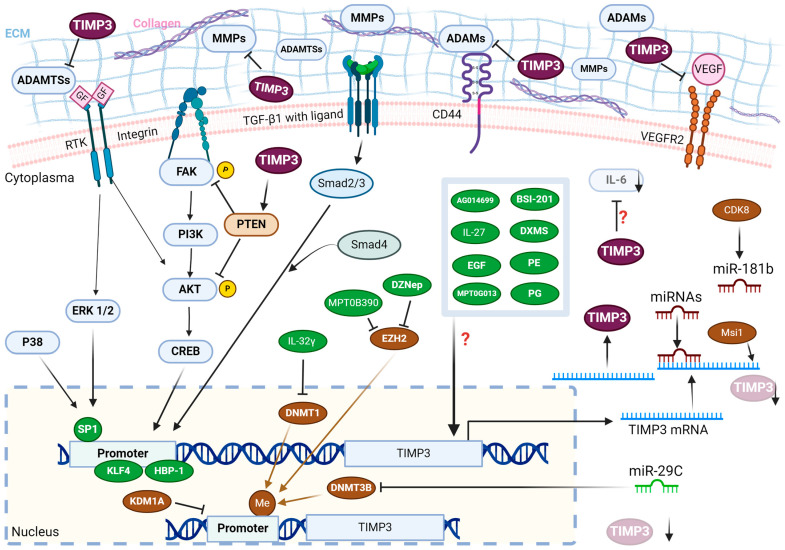
The regulation and the effects of TIMP3 in cancer cells. The molecules with the green icon increased the expression levels of TIMP3, while the molecules with the brown icon decreased the levels. ADAMs, A disintegrin and metalloproteases; ADAMTSs, ADAM with thrombospondin motifs; CDK8, Cyclin-dependent kinase 8; DXMS, dexamethasone; DZNep, 3-Deazaneplanocin A; ECM, extracellular matrix; EGF, epidermal growth factor; EZH2, enhancer of zeste homolog 2; GF, growth factor; HBP1, high mobility group (HMG) box-containing protein 1; IL-, interleukin; KLF4, Krüppel-like factor 4; MMPs, matrix metalloproteinases; Msi1, musashi1; PG, physcion 8-O-β-glucopyranoside; PE, phorbol ester; PTEN, phosphatase and tensin homolog; RTK, receptor tyrosine kinases; TGF-β1, transforming growth factor β-1; TIMP3, tissue inhibitor of metalloproteinases-3; VEGF, vascular endothelial growth factor; VEGFR2, VEGF receptor 2. The symbol “**↑**” indicates the signal transition or binding, and the symbol “**⊥**” shows the blockage of binding or activation. The symbol “?” means the regulatory mechanism is unknown. Created with BioRender.com (accessed on 10 March 2023).

**Table 1 ijms-25-03191-t001:** List of molecules involved in the upregulation of TIMP3 in various cancers.

Reference(s)	Cells	Molecules	Effect
[45]	Colon cancer mice model	MPT0G013	Reduced tumour growth, metastasis, and angiogenesis
[46]	Colon cancer cells and mice model	MPT0B390	Reduced tumour growth and metastasis; increased apoptotic population
[44]	Prostate cancer cells	IL-27	Anti-angiogenic effect
[47]	Liver cancer cells	DZNep	Reduced cell proliferation; increased total apoptosis
[48]	Liver cancer cells	AG014699	Cell proliferation and migration reduction; increased apoptotic population
		BSI-201	
[51]	Lung cancer cells and mice model	KLF4	Decreased cell migration and proliferation
[52]	Lung cancer cells and mice model	IL-32γ	Decreased cell proliferation; increased apoptotic population
[50]	Breast cancer cells	miR-29c	Decreased cell proliferation, migration, and invasion

DZNep, 3-Deazaneplanocin A; IL-, Interleukin-; KLF4, Krüppel-like factor 4.

**Table 2 ijms-25-03191-t002:** List of molecules involved in the downregulation of TIMP3 in various cancers.

Reference(s)	Cells	Molecules	Effect
[56]	Colorectal cancer cells	CDK8	Increased miR-181b and colon cancer growth in the liver
[65]	NSCLC cells	KDM1A	Increased cell proliferation, migration, and invasion
[66]	NSCLC (SCC)	miR-17	Linked to the angiogenesis
[66]	NSCLC (SCC)	miR-20a	ECM deregulation
[67]	Breast cancer cells	miR-21	Increased cell invasion
[68]	Cervical cancer cells	miR-21	Increased MMP2 and MMP9
[69]	Endometrial cancer cells	miR-103	Increased cell growth and invasion
[70]	Liver cancer cells and mice model	miR-181b	Increased cell proliferation, migration, invasion, and resistance to doxorubicin Increased cell proliferation
[71]	Endometriosis cell line Endometriosis-associated ovarian cancer	miR-191	Increased cell proliferation and invasion
[72]	Lung cancer cells	miR-197-3p	Increased angiogenesis
[73]	Oral cancer cells	miR-221	Resistance to Adriamycin
[74]	Oral cancer cells	miR-221	Resistance to doxorubicin
[75]	Thyroid cancer	miR-221/222	Increased aggressiveness
[76]	Cervical cancer cells	miR-221/222	Increased proliferation, migration, and invasion
[77]	Breast cancer cells	miR-221/222	Decreased sensitivity to tamoxifen
[78]	Bone cancer cells and model	miR-222-3p	Increased proliferation and invasion
[79]	Oesophagal cancer cells	miR-373	Increased migration and invasion
[80]	Colorectal cancer cells and mice model	miR-937-5p	Increased proliferation, migration, invasion, and angiogenesis
[81]	Cervical cancer cells	miR-G-10	Increased migration and invasion

CDK8, Cyclin-dependent kinase 8; ECM, extracellular matrix; MMP, matrix metalloproteinases; NSCLC, non-small cell lung cancer; SCC, squamous cell carcinoma.

## Data Availability

Not applicable.

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
