# Peer review of "Tissue Inhibitor of Metalloproteinase 3: Unravelling Its Biological Function and Significance in Oncology"

_ijms, 2024, doi:10.3390/ijms25063191_

Round 1

Reviewer 1 Report

Comments and Suggestions for Authors

The main purpose of the work was correctly indicated in the manuscript by the Authors:

“This review provides an overview of TIMP3, from its biological function to its effects on various cancers. To understand the role of TIMP3 in oncology, including gynecological cancers, the biological function of TIMP3 and its effects on carcinogenesis or cancer progression through the regulation of TIMP3 itself or its related molecules were reviewed and discussed in this study”

1. The Authors described the role of TIMP3 in several types of cancer - why these? TIMP3 is also involved in the pathogenesis of other types of cancer. Please also describe the role of TIMP3 in the pathogenesis of other types of cancer, e.g. hepatocellular carcinoma, colorectal cancer and thyroid cancer.

2. Manuscript should be enriched with more figures to make it more attractive to Readers.

3.     Please unify the fonts used in the tables.

4.     Language polishing is suggested.

5.     Section “Conclusion and Perspectives” was correctly written by the Authors, it is a good summary of the topic presented in the review article. Additionally, the summary includes a figure for better understanding the biological functions of TIMP3.

6.    The references were selected correctly

Comments on the Quality of English Language

Language polishing is suggested.

Author Response

Comments and Suggestions for Authors

The main purpose of the work was correctly indicated in the manuscript by the Authors:

“This review provides an overview of TIMP3, from its biological function to its effects on various cancers. To understand the role of TIMP3 in oncology, including gynecological cancers, the biological function of TIMP3 and its effects on carcinogenesis or cancer progression through the regulation of TIMP3 itself or its related molecules were reviewed and discussed in this study” Bacterial species names should be in italics. Unfortunately, I have not counted one instance where the name was italicized.

  1. The Authors described the role of TIMP3 in several types of cancer - why these? TIMP3 is also involved in the pathogenesis of other types of cancer. Please also describe the role of TIMP3 in the pathogenesis of other types of cancer, e.g. hepatocellular carcinoma, colorectal cancer and thyroid cancer.

Response: Thanks for your valuable suggestions. We have described the role of TIMP3 in hepatocellular carcinoma, colorectal cancer and thyroid cancer in section 4.

  1. Manuscript should be enriched with more figures to make it more attractive to Readers.

Response: We thank the reviewer for this insightful comment. We have added a figure regarding the TIMP3 regulation by different molecules in different cancers.

  1. Please unify the fonts used in the tables.

Response: We have unified the fonts.

  1. Language polishing is suggested.

Response: We thank the reviewer for the helpful suggestions. We have submitted this manuscript to a professional language editing service to improve the quality.

  1. Section “Conclusion and Perspectives” was correctly written by the Authors, it is a good summary of the topic presented in the review article. Additionally, the summary includes a figure for better understanding the biological functions of TIMP3.

Response: Thank you for your affirmation and encouragement.

  1. The references were selected correctly.

Response: Thank you for your affirmation and encouragement.

Reviewer 2 Report

Comments and Suggestions for Authors

The authors aimed to summarize the functions and biology of TIMP3 and its relevance in cancer. Abstract meet their aim; however, the body of this manuscript contain several non-related and repeated parts.

Recommendations:

·        Firstly, TIMP3 can be differently regulated and it can have variable functions in many diseases. Whether you want to evaluate the significance of TIMP3 in oncology, I recommend omitting other diseases and focusing only on studies performed in normal and cancer cells.

·        Introduction – Incorporate more strictly the association between list of therapeutic molecules in different cancers and the last paragraph about TIMP3.

·        Paragraph 2., Tables 1. and 2. - Omit the results from other diseases; the results in different pathologies as probably do not indicate the TIMP3 functions in carcinogenesis.

·        Paragraph 3.1. Upregulation of TIMP3 – Data about downregulation of TIMP3 were also included.

·        Tables 2 and 3. – Do not present the results, which were only discussed, not experimentally validated.

·        Paragraph 4. – TIMP3 in oncology – please specify the title. Several results were presented previously in the text or tables.

·        Paragraph 4.2. – In addition to head and neck cancer, several other cancers were presented.

·        Table 4. – Define low and high level of TIMP3. Any cut-off?

·        Paragraph 5. – Breast cancer is not gynecologic cancer.

·        Table 5. Effect of changed TIMP3 expression is cancer cells compared to normal cells is carcinogenesis ??? It is logical; this column is do not necessary.

·        Tables - Arrange the order of columns more logically.

·        Associations of TIMP3 changes with clinical parameters of cancer patients could be summarized in extra paragraph.

·        Figure 1. -  If you correctly include only results from cancer studies, the title need to be “The regulation and the effects of TIMP3 in cancer cells. “,  

·        Conclusions and Perspectives – The authors discussed the stimulation of TIMP3 as a therapeutic strategy in cancers with downregulated TIMP3, but what about the cases with upregulated TIMP3?

This manuscript need clear structure and according it re-writing with more strict respect to the aim. Therefore, I recommend the major revision.

Comments on the Quality of English Language

Moderate editing.

Author Response

The authors aimed to summarize the functions and biology of TIMP3 and its relevance in cancer. Abstract meet their aim; however, the body of this manuscript contain several non-related and repeated parts.

Recommendations:

  1. Firstly, TIMP3 can be differently regulated and it can have variable functions in many diseases. Whether you want to evaluate the significance of TIMP3 in oncology, I recommend omitting other diseases and focusing only on studies performed in normal and cancer cells.

Response: We thank the reviewer for the helpful suggestions. We have revised the manuscript and focused on studies in normal and cancer cells.

  1. Introduction – Incorporate more strictly the association between list of therapeutic molecules in different cancers and the last paragraph about TIMP

Response: We thank the reviewer for the helpful suggestions. We have revised the manuscript.

  1. Paragraph 2., Tables 1. and 2. - Omit the results from other diseases; the results in different pathologies as probably do not indicate the TIMP3 functions in carcinogenesis.

Response: We thank the reviewer for the helpful suggestions. We have deleted Table 1, which contained information about other diseases, and omitted those diseases from the table.

  1. Paragraph 3.1. Upregulation of TIMP3 – Data about downregulation of TIMP3 were also included.

Response: We thank the reviewer for pointing this out. We have removed the data.

  1. Tables 2 and 3. – Do not present the results, which were only discussed, not experimentally validated.

Response: We thank the reviewer for this insightful comment. We have removed the results.

  1. Paragraph 4. – TIMP3 in oncology – please specify the title. Several results were presented previously in the text or tables.

Response: Thank you for your valuable suggestion. We have revised the title to “TIMP3 in non-gynecological cancers”.

  1. Paragraph 4.2. – In addition to head and neck cancer, several other cancers were presented.

Response: Thank you for your valuable suggestion. We have added the content regarding different cancer types.

  1. Table 4. – Define low and high level of TIMP3. Any cut-off?

Response: Su et al. [39] reported the number of ELISA results and showed the cutoff value. Most of the studies presented the Q-PCR or immunohistochemistry results for TIMP3 and did not show definable value for cut-off.

  1. Paragraph 5. – Breast cancer is not gynecologic cancer.

Response: The content regarding breast cancer has been moved to the non-gynecological cancer section.

  1. Table 5. Effect of changed TIMP3 expression is cancer cells compared to normal cells is carcinogenesis ??? It is logical; this column is do not necessary.

Response: Thank you for the suggestions. The “carcinogenesis” column has been removed in Tables.

  1. Tables - Arrange the order of columns more logically.

Response: Thanks for reviewer’s advice. We have rearranged the columns in tables.

  1. Associations of TIMP3 changes with clinical parameters of cancer patients could be summarized in extra paragraph.

Response: Thanks for your valuable suggestions. We have summarized the TIMP3 changes with clinical parameters of cancer patients.

  1. Figure 1. -  If you correctly include only results from cancer studies, the title need to be “The regulation and the effects of TIMP3 in cancer cells. “,  

Response: Thanks for the valuable suggestions. We have revised the original Figure 1 (It is removed to Figure 2 in the revised manuscript) and the title has been revised.

  1. Conclusions and Perspectives – The authors discussed the stimulation of TIMP3 as a therapeutic strategy in cancers with downregulated TIMP3, but what about the cases with upregulated TIMP3?

Response: Most of the studies focused on the strategies for upregulating TIMP3 in cancer cells or patients with cancer because TIMP3 has been considered a tumor suppressor. Regarding the cases with upregulated TIMP3, the detailed mechanisms should be studied and further validated.

  1. This manuscript need clear structure and according it re-writing with more strict respect to the aim. Therefore, I recommend the major revision.

Response: We thank the reviewer for the valuable suggestions. We have reorganized and revised the manuscript. We have moved the same cancer section together and extended the content for non-gynecological cancers.

Round 2

Reviewer 1 Report

Comments and Suggestions for Authors

The article has been corrected in accordance with the comments and may be published in its current version.